

# Global analysis of Twitter communication in corporate social responsibility area: sustainability, climate change, and waste management

Lucie Kvasničková Stanislavská[1], Ladislav Pilař[1], Xhesilda Vogli[1], Tomas Hlavsa[2], Kateřina Kuralová[1], Abby Feenstra[3], Lucie Pilařová[1], Richard Hartman[1] and Joanna Rosak-Szyrocka[4]

[1] Department of Management, Faculty of Economics and Management, Czech University of Life Sciences Prague, Prague, Czech Republic
[2] Department of Statistics, Faculty of Economics and Management, Czech University of Life Sciences Prague, Prague, Czech Republic
[3] Management Club at the Faculty of Economics and Management, Czech University of Life Sciences Prague, Prague, Czech Republic
[4] Department of Production Engineering and Safety, Faculty of Management, Czestochowa University of Technology, Czestochowa, Poland

## ABSTRACT

Many people now consider social media to be an integral part of their daily routines, which has enabled companies to implement successful corporate social responsibility campaigns through these platforms. The direct interaction with stakeholders offered by social media helps companies to build understanding, trust, and their image. The aim of this study was to identify key topics and trends communicated in connection with corporate social responsibility on the Twitter social network from 2017 to 2022. Analysis of 520,638 tweets by 168,134 unique users identified a predominance of environment-related topics: Sustainability, Climate Change, and Waste management. However, Charity remains the largest single topic. Based on the trend analysis, the areas of ESG, Social Impact, and Charity were identified as growth areas in communication, while Green and Philanthropy, on the other hand, were identified as decreasing.

## INTRODUCTION

Corporate social responsibility (CSR) initiatives have become a priority for business organisations around the world in recent years (*Chae & Park, 2018*; *Suttipun et al., 2021*; *Wang et al., 2016*), as a result of increased competition among businesses to obtain customers by building their image in society through socially responsible activities such as support of employee's well-being, charity's programs, cooperation with the local community, or environmental initiatives (*Rizwana, Atif & Cheema, 2012*; *Tan, Rasoolimanesh & Manickam, 2022*; *Yang, Ngai & Lu, 2020*).

Corresponding author
Ladislav Pilař, pilarl@pef.czu.cz

As the global emphasis on economic and environmental sustainability grows, firms are compelled to declare their corporate social responsibility (*Chen, Hung & Wang, 2018*; *Nguyen, Nguyen & Hoai, 2021*). Regulators, investors, and other stakeholders, particularly in emerging economies, are interested in this trend toward mandating CSR disclosure (*Wu, Zhang & Wei, 2021*).

The exponential growth of social media (SM) has evolved in a transformation in the communication environment in which stakeholders as users are able to publish and share their opinions about businesses and their products (*D'Andrea et al., 2019*). This is a particularly important activity as recent studies have shown the benefits of CSR communication with stakeholders (*Bartikowski & Berens, 2021*; *Husnaini, Sasanti & Cahyaningtyas, 2018*; *Viererbl & Koch, 2022*; *Withisuphakorn, 2018*). This means that the board of directors, attorneys, the HR department, sales department, customer service department, marketing and PR department, and other divisions can use social media in practically every aspect of an organisation's operation (*Edyta, Rizun & Paulina, 2016*). In the context of the increasing popularity of social media, which now has over 4.6 billion users, social media presents significant conceptual and theoretical challenges to a CSR monitoring and reporting literature that has previously mostly concentrated on one-way, yearly communication. Disclosure with CSR reports has been impacted by social media, which provides a more dynamic, participatory public forum (*Neu et al., 2020*).

Twitter is one of the most popular social networking platforms, used by 329 million monthly active users worldwide (*Dixon, 2022*). The rise in popularity of Twitter, as well as people's acceptance of it, has created an opportunity for more research on this social media medium (*Manuel, 2015*). In the last decade, Twitter has established itself as an essential research platform, which has been used in more than ten thousand research articles (*Antonakaki, Fragopoulou & Ioannidis, 2021*), because Twitter allows for obtaining a large amount of real-time textual data of public opinion on the monitored issue (*Gaytan Camarillo et al., 2021*), tracking trends (*Vargas et al., 2021*), and analysing sentiment (*Park, Choi & Jung, 2022*). In the field of CSR, Twitter was used as a data source, for example, in a study by *Dong & Rim (2019)*, which identified and interpreted the non-profit organisations communication strategies used when they are communicating on Twitter about their CSR collaborations; a study by *Okazaki et al. (2020)* analysed the level of interaction in eight brands' CSR Twitter dialogues and investigated the reason for the observed level of interaction; or in the study by *Ali, Frynas & Mahmood (2017)*, where the author identified the determinants of corporate social media accounts for CSR disclosures. However, none of the previous research dealt with the topic structure of CSR and the sentiment that the communicated topics evoke. Similarly, previous studies have yet to deal with trends in CSR communication on Twitter. Based on these findings, the aim of this study is to investigate the use of hashtags, the topic structure, sentiments, and trends in corporate social responsibility (CSR)-related discussions on Twitter.

To achieve the research aim, this study will address the following research questions:

1) What are the most frequently used hashtags in tweets related to CSR?
2) What is the topic structure of CSR-related discussions on Twitter?

3) What sentiments are expressed in CSR-related topics on Twitter?

4) What are the trends in connection with CSR communicated on Twitter?

This study used a social media data analysis with a focus on hashtags and tweets to identify key characteristics of CSR communication on the Twitter social network worldwide. This article contributes to the literature in several ways. First, some previous studies have focused on the social media data analysis of CSR communication at the national (*Amin, Mohamed & Elragal, 2021*), or analysing selected companies (*Maiorescu-Murphy, 2022*; *Okazaki et al., 2020*); this article offers a comprehensive view and offers new findings and insights into CSR communication on Twitter worldwide. When examining the detection of characteristics of CSR communication on Twitter, the article uses an automated machine learning approach to automatically analyse content in tweets instead of using the manual coding techniques commonly used in mainstream CSR communication research. From a methodological point of view, this article represents a new research approach to the analysis of CSR communication on Twitter. The study also expands the current knowledge by identifying dominant topics communicated on Twitter, identifying the sentiment that the communicated topics evoke, and identifying the trend of individual hashtags in connection with the hashtag #CSR independently of the trend of the entire topic #CSR.

## Theoretical background

According to *Hoffman (2007)* the concept of corporate social responsibility dates back to the 1920s, where during this period, there was a shift away from the individual ethic required for westward expansion, and a corresponding rise in the importance of the social ethic necessary for achieving industrial harmony. Throughout the years it has been referred to by a variety of acronyms, including corporate sustainability, voluntary projects, philanthropy, and even just social responsibility (*Wang et al., 2016*). Several decades later, the concept had become more specific, with (*Chonko & Hunt, 1985*) arguing that the "social responsibility" of corporations needed to explicitly name social and environmental interests in particular, *Hopkins (1997)* stating that social responsibility meant the obligations a company had to all of its stakeholders, and *Singhapakdi et al. (1994)* linking social responsibility to a corporation's image, saying companies needed to improve their public brand by implementing actions beyond just philanthropic giving.

Today, *Li et al. (2021)* argue that the most fundamental definition of corporate social responsibility that has stood the test of time is the one developed by *Elkington (1994)*, defining corporate social responsibility as a company's "triple bottom line," *i.e.*, instead of only being motivated by economic gains, a company also needs to take into account social and environmental perspectives.

### CSR reporting

Communication CSR activities are undoubtedly part of the CSR strategy (*Kent & Taylor, 2016*; *Ting, 2021*). In the last few years, the number of firms that have built standardised reports and governance systems to evaluate, assess, drive, and communicate sustainability activities has risen rapidly (*Eccles, Ioannou & Serafeim, 2014*). Prior studies (*Brochet,*

*Loumioti & Serafeim, 2012*; *Capurro, 2005*) support the fact that companies with a high level of sustainability (economic viability, environmental protection, and social equity) are more likely to have developed stakeholder engagement mechanisms, to be more long-term oriented, and to monitor and disclose non-financial data more frequently. *Arvidsson (2010)* and *Nielsen & Thomsen (2012)* found that company communications around corporate social responsibility could actually counter negative publicity around an organisation by altering public perception. Research has found a number of additional positive benefits associated with a company disclosing its corporate social responsibility strategy: to differentiate themselves from its competitors (*Porter & Kramer, 2006*); to increase the possibility of consumers purchasing their products (*Lee & Shin, 2010*; *Wang & Korschun, 2015*); to help consumers identify more with the company (*Tsai et al., 2015*; *Wang & Korschun, 2015*); to improve the company's image and transparency, as well as enhancing investor trust in its investment decisions (*Yusoff, Mohamad & Darus, 2013*).

*Wehmeier & Schultz (2011)* posit that any corporate social responsibility storytelling needs to be based on the values of the corporation, particularly values that stakeholders share and experience themselves, and that these values can be communicated in a way that emphasises the moral implications of the company's actions. Additionally, corporate social responsibility stories are best received when their narratives focus not just on the current actions a company is taking, but on aspirations the company has for initiatives and programmes to effect a change in the future (*Castelló, Morsing & Schultz, 2013*). Further research indicates that for corporate social responsibility communication to be effective, it needs to focus on credibility and the use of well-respected sources, incorporating third-party endorsements from reliable organisations such as non-profits and non-governmental organisations, streamlining media and communication channels, and involving stakeholders themselves (*Maignan & Ferrell, 2001*; *Morsing, Schultz & Nielsen, 2008*; *Pomering & Dolnicar, 2009*; *Schlegelmilch & Pollach, 2005*). Involving stakeholders can be as simple as tailoring corporate social responsibility communications content to include events and examples considered relevant by stakeholders (*Graham Spickett-Jones, Kitchen & Reast, 2004*; *Schlegelmilch & Pollach, 2005*), though it can also involve directly communicating with stakeholders themselves *via* social media interactions.

### CSR reporting and social media

General information may be communicated through a variety of communication instruments and channels. A company can promote its CSR initiatives through official documents such as an annual corporate social responsibility report, brochures, websites, advertising, pamphlets, product packaging, television spots, cause-related marketing, and organising dialogues and other events emphasising corporate social responsibility (*Lodhia, 2004*, *2006*; *Morsing, Schultz & Nielsen, 2008*).

Despite the ubiquity of annual reports as a means for corporate social responsibility communication, recent research may indicate that such reports are gradually becoming less effective and impactful as a communication tool (*Aikat, 2000*; *Saat & Selamat, 2014*). Communicating with stakeholders *via* social networking sites has become an increasingly common practice in the corporate world (*Aichner et al., 2021*; *Araujo, Neijens &*

*Vliegenthart, 2015*; *Cartwright, Liu & Raddats, 2021*; *Dwivedi et al., 2021*; *Okazaki et al., 2020*; *Suárez-Rico, Gómez-Villegas & García-Benau, 2018*). Because of the inherently interactive nature of social networking sites, these platforms are one of the best mediums for creating an open dialogue between companies and stakeholders, one of the aforementioned highly effective methods for engaging public interest in corporate social responsibility activities (*Araujo, Neijens & Vliegenthart, 2015*; *Lee, Oh & Kim, 2013*; *López, Sicilia & Moyeda-Carabaza, 2017*; *Ruehl & Ingenhoff, 2015*). Social media dialogue about CSR helps organisations build understanding, trust, and social capital between organisations and the public (*Kent & Taylor, 2016*).

Around 4.6 billion users are currently active on social networks worldwide (*Dixon, 2022*). In the context of classic mass media such as radio, television, and newspapers, these users are not merely passive recipients but create an active and passive digital footprint (*Deeva, 2019*). Given the fact that in terms of the world's total population (7.9 billion people), social networks are used by around 58% of the population (*Worldometer, 2022*), analysis of this big data is important in understanding the attitudes, experiences, and behaviours of the individual users on these platforms (*Childers, Lemon & Hoy, 2019*; *Pilar et al., 2017*; *de Veirman, Cauberghe & Hudders, 2017*; *Zhang et al., 2020*).

The huge popularity of these social networks is obvious to businesses, which use these platforms to communicate with the wider public (*Pardo, Pagani & Savinien, 2022*). *Tao & Wilson (2015)* report that Twitter is a communication priority for many organisations. Social media interactions with a company on Twitter are more specifically linked to brand identification and loyalty (*López, Sicilia & Moyeda-Carabaza, 2017*). Since Twitter is a well-established social media platform for people to connect with others who share common goals and interests (*Etter, 2014*), it is a logical medium for companies to target for corporate social responsibility communications, which are aspirational and value-based in nature. The benefits of engaging on Twitter for corporations are twofold: tweeting about corporate social responsibility is a more effective way to communicate CSR initiatives than other communication methods, and tweets about corporate social responsibility tend to be viewed more favourably than other company tweets (*Lee, Oh & Kim, 2013*) and may lead to an increase in follower count and engagement with the company's social media account (*Araujo & Kollat, 2018*).

## MATERIALS AND METHODS

The data analysis was based on the extended SMAHR framework (*Pilař et al., 2021*). The data analysis process based on SMAHR framework consisted of four steps (Fig. 1):

1) **Data acquisition**:
   The aim of this step was to collect data from the Twitter social network that contain the hashtag #csr on the Twitter social network from January 1st, 2017 to April 30th, 2022, using the Twitter API.
   The Twitter API was used to obtain the data (*Platform Developer, 2022*).
   Tweets were downloaded from the Twitter social network using the Twitter API v2

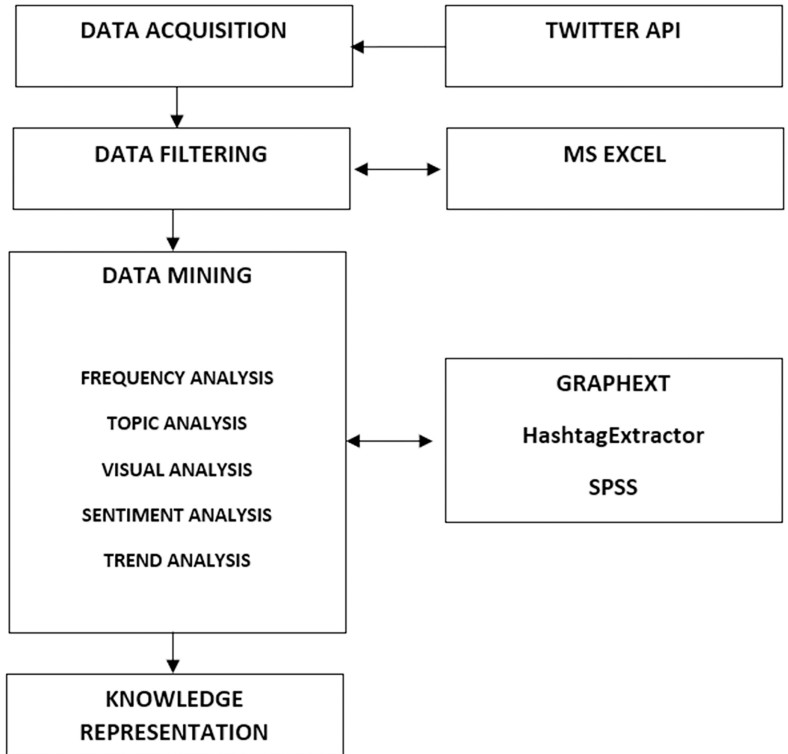

**Figure 1 Four steps of social media analysis based on the framework for hashtag research (SMAHR).**

(*Platform Developer, 2022*). The Tractor software was used for this purpose (*Tractor, 2023*). The software downloaded tweets that included the hashtag #csr. During that timeframe, 520,638 Tweets of 168,134 unique Twitter users were recorder. This dataset includes all tweets with the hashtag #csr that users posted to the Twitter social network during the observation period.

2) **Data filtering**: In the next stage, it was necessary to filter messages that referred to the topic of corporate social responsibility. An example is the computer game "CSR Racing". Filtering was perform manually using the MS Excel program *via* the content filtering function.

3) **Data mining**:
   The aim of this step is to extract useful and relevant information from the vast amounts of data generated by users on social media platforms.
   The network was described using the techniques listed below:

   a. Frequency: A metric called frequency describes how frequently a hashtag appears in a network. The aim of this step was to identify the most used hashtags. Through Hashtag extractor software (*Pilař et al., 2021*), all hashtags were extracted from individual Tweets. The calculation of the number of individual hashtags was performed by importing into Gephi 0.9.2 software (*Bastian, Heymann & Jacomy, 2009*).

b. Topic analysis:

Topic analysis is a method used to identify the main topics or themes that are being communicated within a large dataset, such as social media posts. In complex networks, such as social media networks, some nodes (*i.e.*, hashtags or words) are more interconnected with each other than with the rest of the network. It is possible to identify topics based on clusters of individual hashtags.

The aim of this step was to identify topic structure of CSR-related discussions on Twitter. Compared to frequency analysis, topic analysis was created based on whole tweets (not only hashtags). For topic analysis, Graphext software was used (*Graphext, 2023*). To analyse the community structure of our network, Graphext utilized a modified version of the Louvain algorithm (*Blondel et al., 2008*). The network was created based on the interconnectedness of individual words in the Tweet. The Louvain algorithm employs an iterative process of allocating nodes to clusters with the aim of optimizing a performance metric known as modularity. This metric gauges the relative density of edges within clusters compared to those between clusters. The number of distinct communities in the dataset was calculated as follows:

$$\Delta Q = \left[ \frac{\sum_{in} + 2k_{i,in}}{2m} - \left( \frac{\sum_{tot} + k_i}{2m} \right)^2 \right] - \left[ \frac{\sum_{in}}{2m} - \left( \frac{\sum_{tot}}{2m} \right)^2 \right],$$

where $\sum_{in}$ is the sum of weighted links inside the community, $\sum_{tot}$ is the total number of weighted connections inside the community, $k_i$ is the total number of weighted links related to community hashtags, and i, $k_{i,in}$ is the total weighted linkages from an individual to community hashtags, and m is the normalization factor, calculated as the total weighted links over the entire graph (*Blondel et al., 2008*).

c. Visual analysis: Network visualization techniques such as force-directed layouts can be used to highlight different aspects of a network, such as the density of connections, or polarization of topics. The aim of this step was to identify the polarity of the identified topics.Based on the ForceAtlas2 layout technique, a two-dimensional graph was generated for visual analysis. An improved version of the ForceAtlas algorithm called ForceAtlas2 focuses on massive networks was used. This approach uses visual representations of smaller samples to identify network communities' intercommunity connections (*Jacomy et al., 2014*). Visual analysis was created using Graphext software (*Graphext, 2023*).

d. Sentiment analysis: The purpose of sentiment analysis on Twitter social media is to determine the general attitude or opinion expressed in tweets about a particular topic. Sentiment analysis is a process of categorizing text into positive, negative, or neutral sentiment based on the context and tone of the language used.

The aim of this step was to identify sentiments, which are expressed in CSR-related topics on Twitter.

In this research, VADER (Valence Aware Dictionary and Sentiment Reasoner) was

used by sentiment analysis. VADER is a Lexicon and Rule-Based Sentiment Analysis Tool (*Hutto & Gilbert, 2014*).

e. Trend analysis: The purpose of trend analysis is to identify and understand patterns and changes over time. The aim of this step was to identify the trends in connection with CSR communicated on Twitter *via* hashtags trend analysis. For trend analysis was used SPSS 21 software. In the field of trend analysis, three following steps were used:

i) Tests for trends in time series: There was assessed general development tendency of all-time series through the linear trend function. Along with it there was developed the null hypothesis, meaning that the slope measured by regression coefficient is not statistically significant. To verify the reject or fail to reject of the null hypothesis, the *t*-test was used. Decisions are made by comparing the maximum first type error (the *p*-value), based on our data, and the first type error of alpha (*Tufféry, 2011*).

ii) Cluster analysis: Time-series clustering is a process that can help us to understand the pattern in fluctuating and large time series data (*Tufféry, 2011*). Time series clustering is an unsupervised learning problem similar to the clustering in other data and variable types, but we also take the time variable into account (*Murtagh & Legendre, 2014*). We mainly perform time series clustering to minimize the data similarity across the clusters and increase the similarity within the cluster. We used agglomerative clustering in our analysis, because we explore the time series, and the number of clusters is not known. We prepared the time series for analysis, checked multicollinearity, and standardized all input values using z-score. Then we calculated the similarity using Euclidian distance and we employed Ward's method as the clustering algorithm. The homogeneity of clusters was measured by R-square and Semipartial R-square (*Tufféry, 2011*). R-square is the proportion of the sum of squares explained by the clusters (between-cluster sum of squares/total sum of squares). The nearer it is to 1, the better the clustering will be. Semipartial R-square measures the loss of the between-cluster sum of squares caused by grouping two clusters together (*Tufféry, 2011*). Thus, the value should be small to imply that we are merging two homogeneous groups.

iii) Chow test: We examined whether two parts of the time series show a different slope. A method commonly method used for this is the Chow test (*Binkley & Young, 2020*), which tests for group effects by comparing the error sum of squares (ESS) from regressions on the individual time series to the ESS from a pooled regression using an *F*-test.
Consider a standard $k$-variable regression model $Y = \alpha + \mathbf{X}\beta$ (*Binkley & Young, 2020*), where $e$ is the usual error term and $k$ includes an intercept. Data is available from two distinct parts, the first and the second section of the time

**Table 1 The 30 most frequently used hashtags related to CSR on Twitter sorted by frequency.**

| No. | Hashtag | Frequency | No. | Hashtag | Frequency |
|-----|---------|-----------|-----|---------|-----------|
| 1 | #csr | 520,638 | 16 | #india | 8,502 |
| 2 | #sustainability | 100,474 | 17 | #socialresponsibility | 8,203 |
| 3 | #esg | 36,347 | 18 | #supplychain | 8,035 |
| 4 | #sgds | 19,860 | 19 | #sustainable | 7,927 |
| 5 | #green | 15,488 | 20 | #marketing | 7,111 |
| 6 | #business | 14,442 | 21 | #climatechange | 6,902 |
| 7 | #socialimpact | 13,429 | 22 | #socenet | 6,691 |
| 8 | #leadership | 12,282 | 23 | #community | 6,298 |
| 9 | #charity | 11,076 | 24 | #marketing | 6,291 |
| 10 | #corporatesocialresponsibility | 10,840 | 25 | #education | 6,278 |
| 11 | #philanthropy | 10,373 | 26 | #impact | 6,219 |
| 12 | #health | 10,276 | 27 | #mentalhealth | 5,934 |
| 13 | #environment | 9,405 | 28 | #colunteering | 5,895 |
| 14 | #corpgov | 9,178 | 29 | #psychology | 5,879 |
| 15 | #ethics | 9,101 | 30 | #innovation | 5,866 |

series. Let us denote the groups as A and B, the interest is in whether the same equation applies to both. There we test the Chow test null hypothesis $\beta A = \beta B$. The Chow test statistic is as follows (*Chow, 1960* in *Binkley & Young, 2020*):

$$F = \frac{ESS_P(ESS_A - ESS_B)}{ESS_A + ESS_B} \times \frac{n_A + n_B - 2k}{k}$$

where the ESS's are the error sum of squares from the regressions. The statistic has an *F*-distribution with $k$ and $nA + nB - 2k$ degrees of freedom. The null hypothesis tested assumes equal regression coefficients.

4) **Knowledge representation**: Knowledge represenation is a procedure that uses visualization tools to explain the findings of data mining. Knowledge representation highlights the synthesis of individual values and outputs from the phase of data evaluation. The aim of this step is to highlight important findings of previous analyses.

## RESULTS AND DISCUSSION

First of all, data from the period 2017–2022 was used to analyse the frequency of hashtags in relation to the #CSR hashtag. The hashtags most posted in connection with #CSR are #sustainability, #esg, #green and #sdgs—see Table 1. According to many authors (*Księżak & Fischbach, 2018*; *Pan, Sinha & Chen, 2021*; *Varyash et al., 2020*), CSR is based on the Triple Bottom Line concept, according to which a business is based on three pillars—profit, people, planet (*Uadiale & Fagbemi, 2012*), *i.e.*, it bears economic, social, and environmental responsibility for its activities (*Braccini & Margherita, 2018*). The results of the hashtag frequency analysis on Twitter show that environment-related hashtags are the

most common. The top 30 most frequent hashtags in CSR communication include #sustainability, #green, #sdgs, #sustainable, #environment and #climatechange.

The World Commission on Environment and Development (*United Nations, 1987*) defines sustainability as utilizing resources to meet the 'needs of the present without compromising future generations' ability to meet their own needs'. The close link between CSR and sustainability has long been confirmed by numerous studies (*Carroll & Shabana, 2010*; *Christensen, Hail & Leuz, 2021*; *Kang et al., 2015*; *Meseguer-Sánchez et al., 2021*). A bibliometric analysis from 2003–2021 conducted by *Sánchez-Teba et al. (2021)* found that the main topics driving CSR research between 2003 and 2021 were sustainability and the environment. The strong link between the concepts is particularly clear in Europe, where for the European Union CSR is the basic means of supporting sustainable development (*Yıldız & Ozerim, 2014*).

Environmental, social and governance (ESG) can be seen as the three non-financial dimensions of business, which take into account the impact on the environment, respect for social values and the quality of company management (*DeGennaro & Barry, 2020*). ESG developed from the CSR concept (*MacNeil & Esser, 2022*) and represents an evaluable result regarding a company's overall sustainability performance (*Polley, 2022*). ESG offers quantitative metrics and thus enables a shift away from qualitative descriptions of CSR (*Cini & Ricci, 2018*). The importance of ESG has risen particularly in recent years, in connection with the new EU directive on corporate sustainability reporting, which from 2024 obliges enterprises with 250 or more employees to report non-financial information (*European Commission, 2021*), thus greatly increasing the number of businesses that will be obliged to file non-financial reports.

The concepts of CSR, ESG and sustainability have much in common; certain (*Fatemi, Glaum & Kaiser, 2018*; *Gillan, Koch & Starks, 2021*) even consider them to be equivalent. However, despite their apparent similarity, these are separate concepts with fundamental differences (*MacNeil & Esser, 2022*).

The hashtag #SDGS, which analysis has shown to be the 5th most frequently posted hashtag in connection with CSR, refers to the 17 Sustainable Development Goals adopted by all UN Member States in 2015. SDGs represent a plan to protect the environment, eradicate extreme poverty, and combat injustice and inequality (*United Nations, 2015*).

## Topic analysis

Topics analysis allows us to better understand the dynamics of the entire communication through the identification of links between individual hashtags. The results of the topic analysis are shown in Table 2.

The largest topic identified is Charity which includes aspects relating to volunteer and philanthropic activities: charity, volunteer, donate, philanthropy, *etc.* These results are in accordance with the research conducted by *Kvasničková Stanislavská et al. (2020)*, which drew on data from Instagram from 2019 to identify philanthropic responsibility as the biggest topic in both developing and developed countries. The same conclusion was also drawn by a study carried out by *Ngai & Singh (2021)*, which analysed the CSR communication of Chinese companies on the Sina Weibo social network. Charity and

**Table 2 Identified topics related to #CSR.** Absolute and percentage size and key words are listed for each topic.

| Topic | Size of the topic | | Key terms |
|---|---|---|---|
| | % | Absolut | |
| Charity | 16.06% | 83,691 | Charity, support, volunteer, help, donate, donation, volunteering, community, philanthropy, nonprofit |
| Company government | 15.78% | 82,203 | Business, company, corporate, impact, work, community, strategy, value, community, good |
| Sustainability | 10.96% | 51,383 | Sustainability, green, sustainable, esg, sdg, report, environmental, business, change, future |
| Education | 7.06% | 36,800 | School, education, child, student, project, skill, young, support, initiative, program |
| Climate change | 5.24% | 27,305 | Climate, energy, carbon, climatechange, renewable, emission, green, sustainability, solar, environment |
| CSR reporting | 3.60% | 18,733 | Corporate social responsibility, csr reporting, social impact, report, company, business, read, impact, esg, responsible |
| Waste | 3.18% | 16,581 | Plastic, water, waste, recycle, food, circular economy, green, sustainability, packaging, reduce |
| Marketing | 2.81% | 14,633 | Brand, purpose, marketing, consumer, customer, business, company, profit, market, social impact |
| Human resources management (HRM) | 2.78% | 14,470 | Job, employee, hire, job opening, workplace, HR, employee management, career, manager, engagement |

philanthropic activities have formed a basic part of the CSR concept since its very beginning (*Carroll, 1991*); our results show that, despite the fact that CSR has seen a major shift towards sustainability, these still have their place in the concept.

The second largest topic identified is Company Government. This topic involves communications relating to the management of a company, such as a strategy, work, business, value, *etc.*

A significant proportion of CSR communication on Twitter is devoted to sustainability. As *Pilař et al. (2019)* state, as evidence of environmental risks mounts up, sustainability is being included into the agendas of legislators and organisations, as well as company objectives. The results of a cluster analysis identified this area in the topics Sustainability, Climate Change and Waste, which together make up 19.38% of CSR communication on Twitter. The dominance of sustainability in CSR communication is also highlighted by a hashtag frequency analysis (Table 1). Our findings can be explained as the importance that environmental issues have taken on in the eyes of the public, as also confirmed by studies by *Connor et al. (2022)*, *Jakučionytė-Skodienė, Krikštolaitis & Liobikienė (2022)* and *Wassmann, Siegrist & Hartmann (2023)*. Companies are recognised as sustainability leaders and use communication about sustainability to attract public attention and to increase pressure on key (*Abeydeera, Tregidga & Kearins, 2016*).

Another topic identified is Education, which includes terms such as school, student, child, project, and skill. Support for education within the framework of CSR activities is a routine part of CSR and is of fundamental significance particularly in developing markets, where companies often use CSR programs to take up the role of the state, where it fails to act itself (*Valente & Crane, 2010*), although they are still used in western countries, too (*del Baldo, 2018*; *Chuah et al., 2022*). According to *Roy, Rao & Zhu (2022)*, investment in education and health care within the framework of CSR programmes gives companies

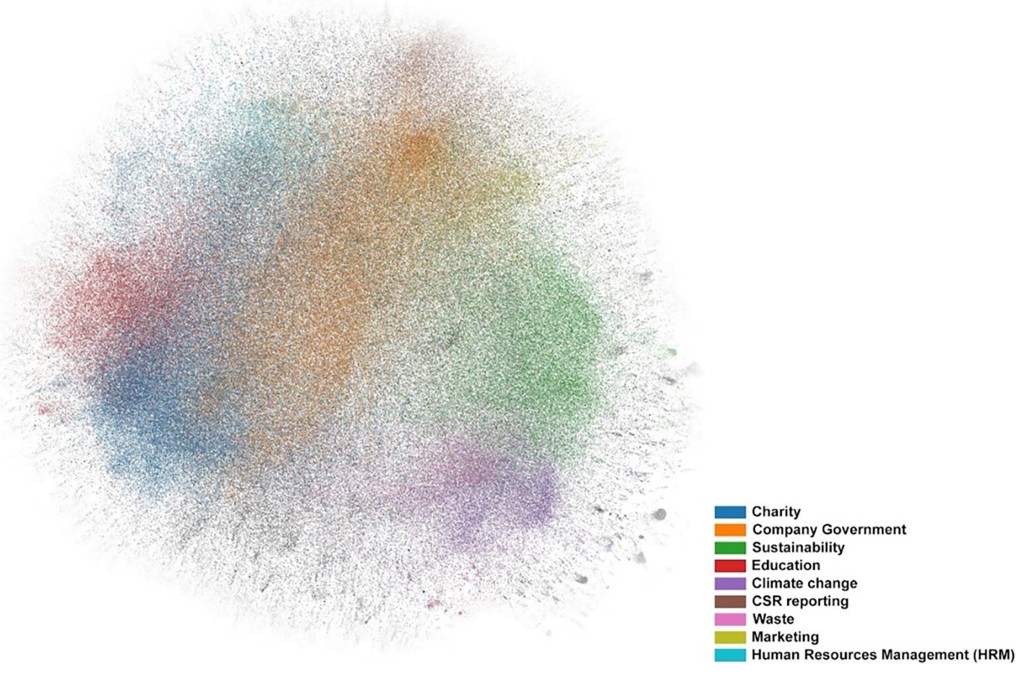

**Figure 2 Visual analysis.** The individual colors represent the extracted clusters in the ForceAtlas2 view.

higher stock market liquidity than investment in sustainability or social justice. A visual analysis (see Fig. 2) has identified that communication in the field of CSR and Education is closely linked to the field of CSR and Charity.

The topic Marketing communicates content aimed at promoting sales, such as purpose, brand, customer, market, *etc*. *Okazaki et al. (2020)* found that certain companies use CSR communication on Twitter solely as another advertising channel and do not make the most of the potential of dialogue with stakeholders for creating value. However, according to *Jahdi & Acikdilli (2009)*, focusing excessively on boosting sales entails the risk of the company losing consumer trust in its CSR communication as a whole.

The last topic identified is Human resources management (HRM), which includes terms such as job, employee, workplace, career, *etc*. HRM can be classed as part of the internal social pillar within the Triple Bottom Line concept defined by *Elkington (1994)*. There are also current studies focusing on CSR and HRM (*Podgorodnichenko, Edgar & Akmal, 2022*; *Stahl et al., 2020*; *Turner et al., 2019*; *Yin et al., 2021*) although previous research analysing CSR on social media had not identified HRM prior to then. This could therefore be a new trend, with the CSR concept gaining strength in HRM.

A visual analysis enabled us to identify the link between Charity and Education together with CSR HRM. Topics polarised from this group are those focusing on the environment and sustainability (Sustainability, Climate Change a Waste). These poles then include the topic Company Government, which overlaps into both groups. These results are in accordance with the partial results achieved by individual studies (*Karagiannis et al., 2022*; *Khan, Lockhart & Bathurst, 2021*; *Ndubuka & Rey-Marmonier, 2019*).

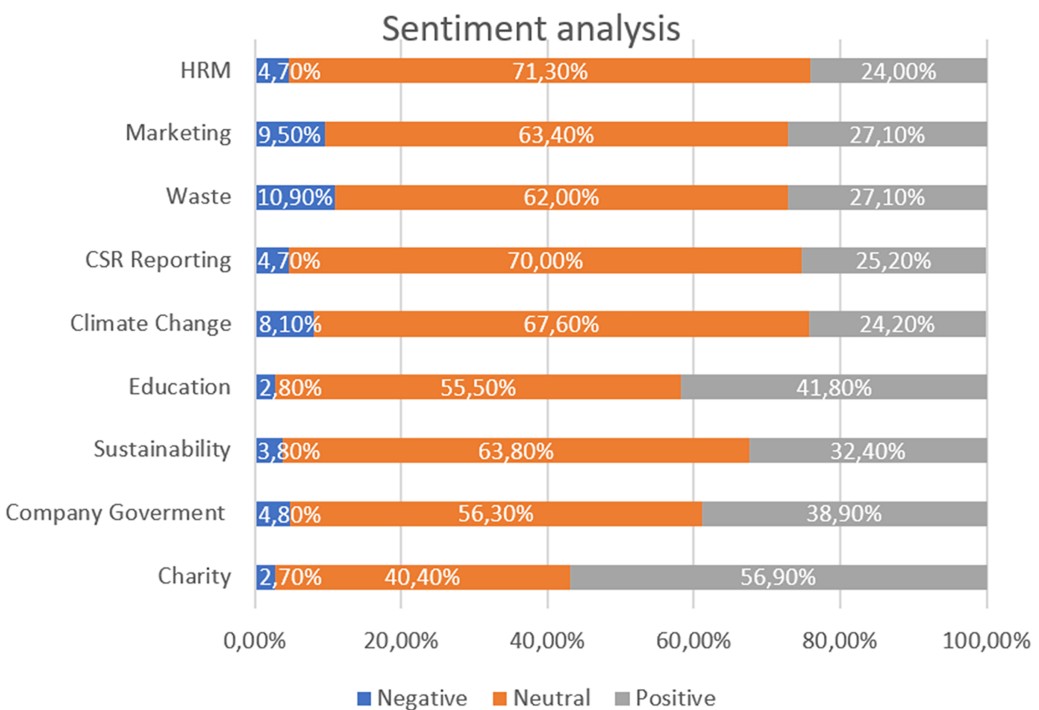

**Figure 3 The percentage representation of individual sentiments in the extracted topics.**

## Sentiment analysis

Sentiment analysis is used to identify the emotions that are represented in the monitored tweets (*Shamoi et al., 2022*). Our results (Fig. 3) shows that the greatest negative sentiment is generated by communication on the topics of Waste (10.9%), Marketing (9.5%), and Climate Change (8.1%). Negative sentiment communicated in relation to the topics of Waste and Climate Change is not necessarily the result of public opposition to communicated CSR activities in these areas but is generally a consequence of negative emotions—such as the sadness and fear that environmental problems elicit in certain parts of the population (*Gago & Sá, 2021*; *Hickman et al., 2021*). In contrast, negative sentiment occurring the Marketing topic is probably related to consumer distrust and their concerns that CSR activities will be misused for companies' marketing purposes and for so-called greenwashing (*Ginder & Byun, 2022*; *Martínez et al., 2020*).

In contrast, the highest positive sentiment can be identified with the topics of Charity (56.9%) and Education (41.8%). Neutral sentiment was most frequently elicited by topics communicated in connection with HRM (71.3%) and CSR Reporting (70%).

The results of the sentiment analysis indicate that companies focusing on topics related to charitable activities or supporting education on Twitter have a high potential to generate positive discussion. On the contrary, posts that deal with environmental topics such as waste management, the effects of climate change, *etc.*, are dangerous to evoke more negative emotions in stakeholders than other topics. It is therefore crucial for companies to

**Table 3 Trend analysis.** The development of the trend of the top nine hashtags.

| Hashtag | Perameter estimate-b | T value | Pr > |t| | Trend |
|---|---|---|---|---|
| Esg | 0.23033 | 14.61 | <0.0001 | Growing |
| SDGs | −0.01389 | −1.3 | 0.1991 | Has not been proven |
| Green | −0.06757 | −4.06 | 0.0001 | Decreasing |
| Business | −0.00312 | −0.56 | 0.5788 | Has not been proven |
| Socialimpact | 0.05222 | 9.57 | 0.0001 | Growing |
| Leadership | 0.00351 | 0.83 | 0.4073 | Has not been proven |
| Charity | 0.01443 | 2.76 | 0.0075 | Growing |
| Philanthropy | −0.02602 | −7.97 | 0.0001 | Decreasing |

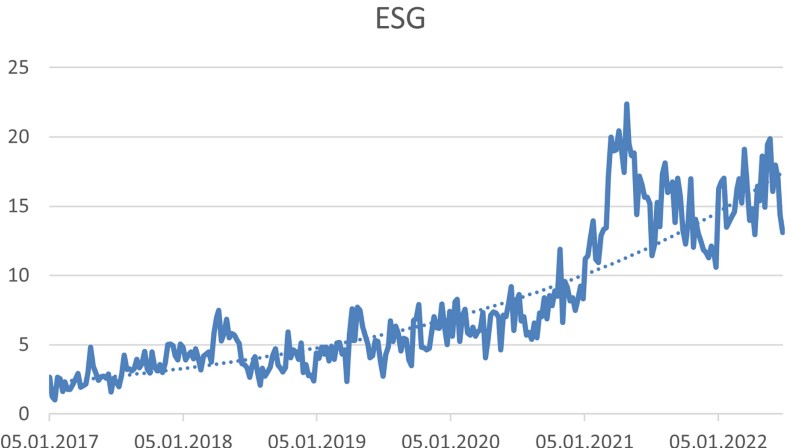

**Figure 4 Trend of the #ESG hashtag on the Twitter social network in connection with CSR.**

be aware of what kind of communication can generate positive reactions and what kind can have a negative impact on customer and public relations when publishing posts.

## Trend analysis

Trend analysis was created for the nine most frequent hashtags that are used in connection with CSR (for more, see Table 3) * Hashtag #corporatesocial was eliminated from the analysis—it is a synonym. These are relative values that are related to the #CSR hashtag. Based on this, it is possible to identify the trend of individual hashtags in connection with the hashtag #CSR independently of the trend of the entire topic #CSR.

## Growing trend

Based on the trend analysis, it was possible to identify three hashtags that have a growing trend. (1) #ESG—this hashtag expresses "Environmental, Social and Corporate Governance", (see Fig. 4), (2) #Socialimpact—this hashtag expresses "significant or positive changes that solve or at least address social injustice and challenges. Businesses or organisations achieve these goals through conscious and deliberate efforts or activities in

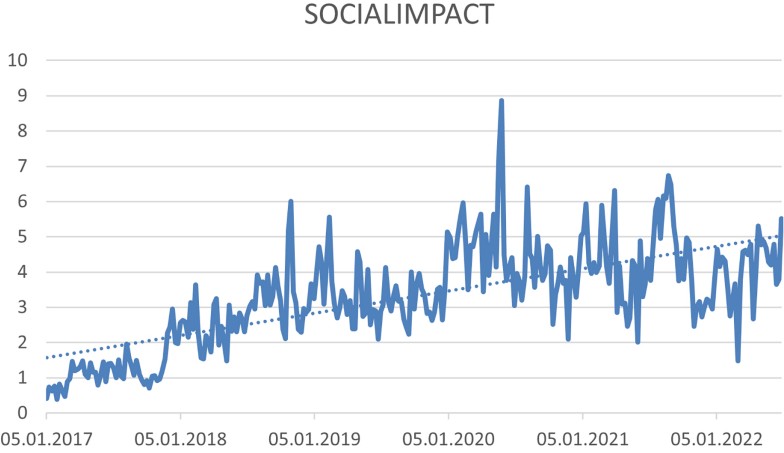

**Figure 5 Trend of the #socialimpact hashtag on the Twitter social network in connection with CSR.**

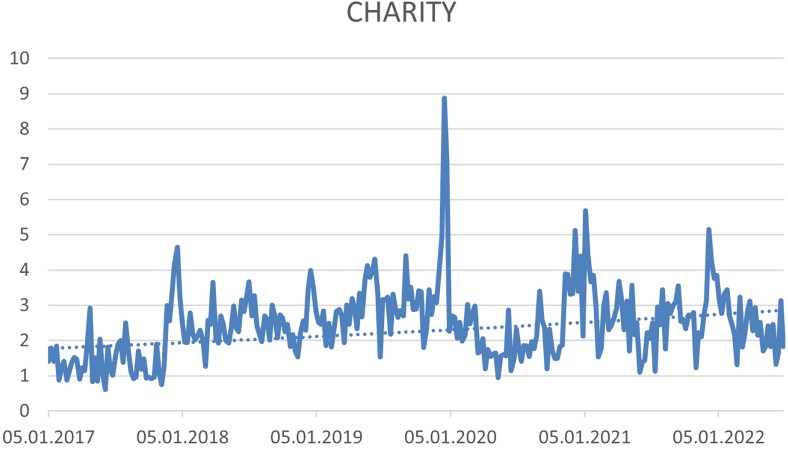

**Figure 6 Trend of the #charity hashtag on the Twitter social network in connection with CSR.**

their operations and administrations" (*Ricee, 2021*) (see Fig. 5), and (3) #Charity—this hashtag expresses "form of money, given freely to people who are in need, for example because they are ill, poor, or have no home" (*English Cambridge Dictionary, 2022*) (see Fig. 6).

Our results confirm the growing popularity of ESG, which is mentioned for example in studies *Zainullin & Zainullina (2021)*, *Clément, Élisabeth & Léo (2022)*, and *Dmuchowski et al. (2023)*. One of the reasons for this trend may be that ESG represents the best available tool for evaluating CSR (*Cini & Ricci, 2018*). According to *Sætra (2021)*, the concept of ESG even replaces the concept of CSR. For these reasons, companies should pay considerable attention to ESG in their social media communications to keep up with current trends.

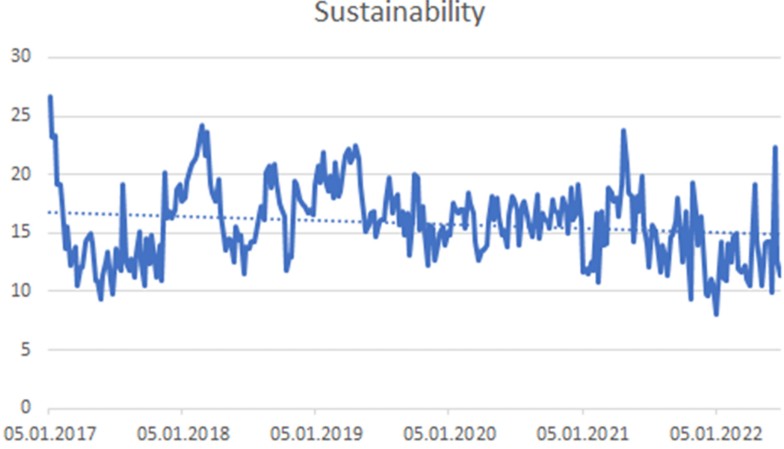

**Figure 7 Trend of the #sustainability hashtag on the Twitter social network in connection with CSR.**

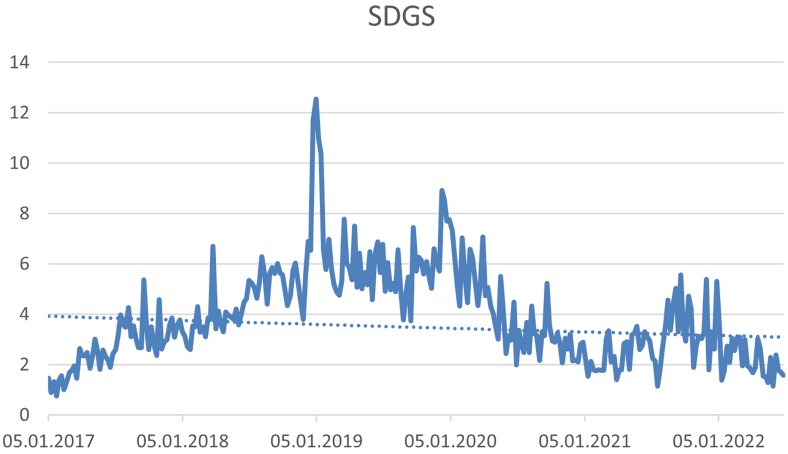

**Figure 8 Trend of the #SGDs hashtag on the Twitter social network in connection with CSR source: own elaboration.**

## The trend has not been proven

The four hashtags have not been proven a trend. (1) #sustainability (see Fig. 7), (2) #sgds—This hashtag expresses the Sustainable Development Goals (see Fig. 8), (3) #business—expressing the connection of CSR with the business environment (see Fig. 9) and (4) #leadership—connecting CSR and Leadership. Leadership is not about leaders, but about CSR processes that need to be supported (*Lythreatis et al., 2021*; *Phillips, Thai & Halim, 2019*) (see Fig. 10).

## Decreasing trend

As decreasing, the following two hashtags (1) #green and (2) #philanthropy, were identified. This is a surprising finding because (*Shiri & Jafari-Sadeghi, 2022*) research identifies the interconnection of Green and Philanthropy with CSR. A possible cause of this result is the shift of CSR towards sustainability.

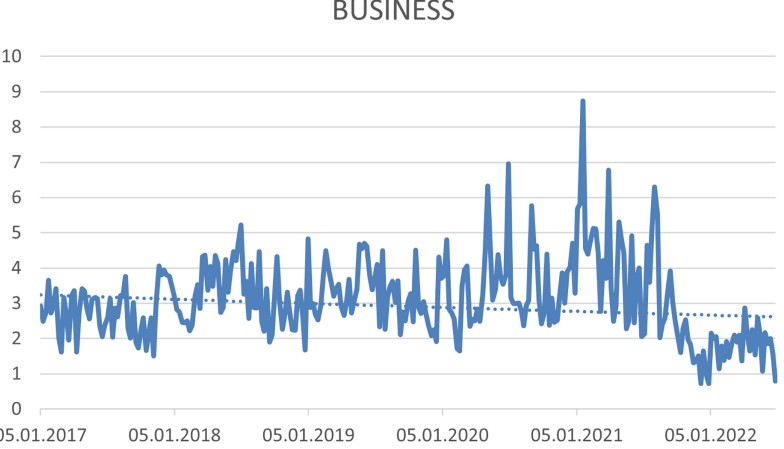

**Figure 9 Trend of the #business hashtag on the Twitter social network in connection with CSR** source: own elaboration.

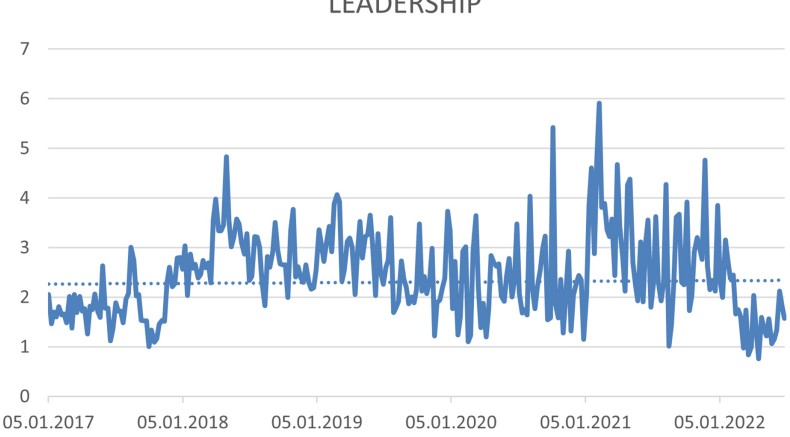

**Figure 10 Trend of the #leadership hashtag on the Twitter social network in connection with CSR** source: own elaboration.

## Cluster analysis of time series

The similarity in the development of the monitored time series was evaluated. The aim was to identify which time series show the most similar development and at the same time to explore whether some time series show a different trend development. Based on this analysis August 2020 was identified as a turning point where ESG values grew more faster than other time series. The most similar are the time series Charity and Leadership, to which the Socialimpact time series joined. Another similar pair is Philanthropy and Business. ESG was the only value that showed significantly different behaviour (see Fig. 11). On the basis of this analysis, the ESG values were performed more detailed to the break using the Chow's test. The development of the time series is then very different for the Sustainability time series, which contained the highest shares of all monitored time series throughout the time series.

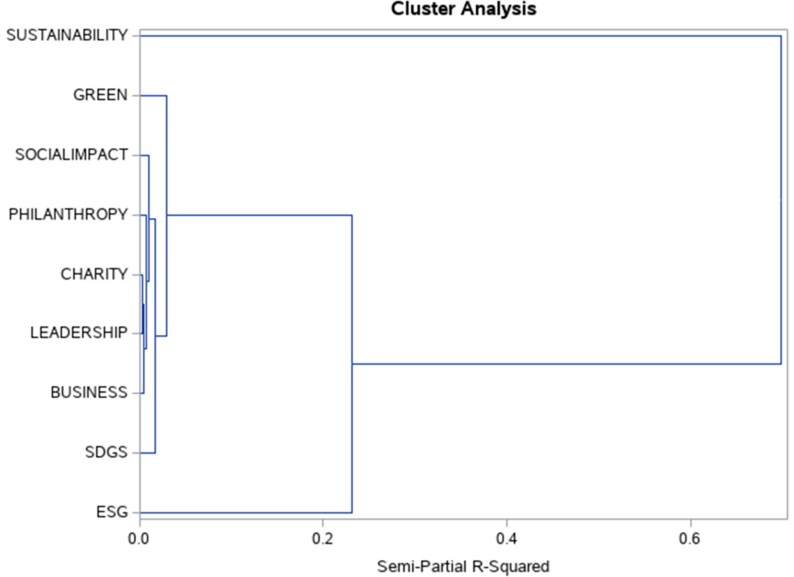

**Figure 11 Dendrogram for all input variables.** The cluster analysis of top nine hashtags connected to CSR.

## ESG time series analysis

### Prerequisite

From August 2020, values grow faster. There is a Chow test that tests the match of the models to the turning point (January 2017–August 2020) with the model from the break to the end (August 2020–September 2022).

### Trend function for the entire time series (January 2017–September 2022)

$y' = 0.041 + 0.23t$

$ESS_P = 380.75$

### Trend function for the first part—(January 2017–August 2020)

$y' = 2.381 + 0.09985t$

$ESS_A = 28.84$

### Trend function for the second part—(August 2020–September 2022)

$y' = -2.59 + 0.29926t$

$ESS_B = 180.09$

## Null hypothesis for Chow test

$H_0$: regression coefficients are equal

$H_1$: non $H_0$

## Calculation of the F-test statistics

$$F = \frac{\dfrac{(380.75 - 28.84 - 180.09)}{2}}{\dfrac{(28.84 + 180.09)}{62}} = 25.49$$

**p-value < 0.001**.

## Null hypothesis can be rejected

We have enough evidence to claim that the slope is in both time series models different. That is the significantly faster growth in the second part of the time series.

## The theoretical and practical implications

The theoretical contributions of this study are manifold. First, as previously mentioned, a holistic view of CSR communication on Twitter needs to be added to the current literature. Although some studies identified some topics in CSR communication on Twitter (*e.g.*, *Chae & Park, 2018*; *Okazaki et al., 2020*), analysing factors in detail has contributed to this study. Second, this study introduces an automated machine learning approach to automatically analyse the content in CSR communication on Twitter instead of using the manual coding techniques commonly used in mainstream CSR communication research, thus, contributing from the methodological point of view.

There are some critical practical implications of this study. As *Hassani & Mosconi (2022)* state, a business can use social media to monitor trends in its business and as a source of competitive intelligence. In the context of our results, companies should emphasize the communication of activities from the environmental area, as this area is the strongest in the structure of the identified topics. Communication in connection with charitable activities within the CSR policy of the company evokes positive sentiment. ESG has clearly become a trend in CSR communication on Twitter in recent years. These results could be used for the strategic management of CSR for planning marketing communication.

## CONCLUSIONS

In the current competitive era, communicating CSR activities is essential to a company's CSR strategy. By social media analysis with a focus on hashtags and tweets, the present study determined the method by which companies communicate CSR on the Twitter social network worldwide. Analysis of this communication has identified a high proportion of environment-related terms in CSR communication. Of the nine topics extrapolated, three relate to the environment: (1) Sustainability, (2) Climate Change, and (3) Waste. The environment also features strongly in CSR communication according to the results of an analysis of the frequency of the individual hashtags, specifically #sustainability, #esg, #green, #sustainable, and #environment. The results also indicate that meeting sustainable development requirements is a high priority for the business sector, which is in line with the results of studies conducted by *Tsalis et al. (2020)* and *Bose & Khan (2022)*. When examining the detection of characteristics of CSR communication on

Twitter, the article uses an automated machine learning approach to automatically analyse content in tweets instead of using the manual coding techniques commonly used in mainstream CSR communication research.

The results of the study show that CSR is a topic that is frequently communicated on Twitter. As pointed out by *Sharma (2019)*, well-communicated CSR campaigns on social media are enormously beneficial to companies. Our results aid enterprises in developing communication strategies that support the company's existence as a socially responsible subject and promote the management of the company's reputation in accordance with the CSR philosophy.

This research brings new challenges for future studies. In connection with the results, in future research, it would be advisable to ascertain whether the topics communicated vary over time.

This research used the Modified Louvain algorithm to extract the individual topics. In follow-up studies, it would be appropriate to make use of other methods, such as Latent Dirichlet Allocation (*Blei, Ng & Jirdan, 2003*), or structural topic models (*Roberts, Stewart & Airoldi, 2016*).

In future studies, expanding the analysis by the local context of individual CSR activities, such as in the study focusing on Italy (*Patuelli et al., 2021*) would provide the identification of differences between individual countries in comparison with the global context and the overall trend of globalization.

Our study identified the trends of unique hashtags in connection with the hashtag #CSR. Further research is needed to learn more about the attributes that lead to these trends.

Limitation of this research, similar to earlier research (*Dong & Rim, 2019*; *Ngai & Singh, 2021*) that focused on the analysis of social networks, this study was focused on a single social network, namely Twitter, which opens up possibilities for further research for other social media, such as Instagram or LinkedIn.

### Funding

This study was supported by the Internal Grant Agency (IGA) of FEM CULS in Prague, registration 2023B0006—Using artificial intelligence to analyze communication on social media. The funders had no role in study design, data collection and analysis, decision to publish, or preparation of the manuscript.

### Grant Disclosures

The following grant information was disclosed by the authors:
Internal Grant Agency (IGA) of FEM CULS in Prague: 2023B0006.

### Competing Interests

The authors declare that they have no competing interests.

## Author Contributions

- Lucie Kvasničková Stanislavská conceived and designed the experiments, analyzed the data, authored or reviewed drafts of the article, and approved the final draft.
- Ladislav Pilař conceived and designed the experiments, analyzed the data, performed the computation work, authored or reviewed drafts of the article, and approved the final draft.
- Xhesilda Vogli analyzed the data, prepared figures and/or tables, and approved the final draft.
- Tomas Hlavsa analyzed the data, performed the computation work, authored or reviewed drafts of the article, and approved the final draft.
- Kateřina Kuralová analyzed the data, prepared figures and/or tables, and approved the final draft.
- Abby Feenstra conceived and designed the experiments, prepared figures and/or tables, and approved the final draft.
- Lucie Pilařová conceived and designed the experiments, prepared figures and/or tables, and approved the final draft.
- Richard Hartman performed the computation work, authored or reviewed drafts of the article, and approved the final draft.
- Joanna Rosak-Szyrocka conceived and designed the experiments, analyzed the data, authored or reviewed drafts of the article, and approved the final draft.

## Data Availability

The data is available at Zenodo: Kvasničková Stanislavská, Lucie, & Pilař, Ladislav. (2023). Tweets IDs related to Twitter communication in the CSR area (2017–2022) [Data set]. Zenodo. https://doi.org/10.5281/zenodo.7799914.

The code (Hashtag extractor) is available from *Pilař et al. (2021)*:

Framework for Social Media Analysis Based on Hashtag Research.

https://doi.org/10.3390/app11083697.

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
