# Peer review of "Global analysis of Twitter communication in corporate social responsibility area: sustainability, climate change, and waste management"

_PeerJ Computer Science, doi:10.7717/peerj-cs.1390_

## Round 0.1 · original submission · Major Revisions

Both reviewers find the paper with potential and interest. However, before considering it for publication, I agree that some polishing is required to improve the research and the manuscript. Both reviewers have identified particular aspects of the methodology as confusing and need to be clarified.
I look forward to a revised version of the manuscript in which the suggestions set forth by the reviewers are incorporated or discussed.

Reviewer 1 ·

Basic reporting

The article presents an appropriate and understandable language, the literature references are correct and the objective of the work and its results are adequately justified and discussed. In addition, it presents an adequate structure and all the results of the hypotheses. What is not correct is the use of acronyms in the text for the first time without indicating what they refer to (line 388 HRM), revise the text.

Experimental design

The research paper is interesting and the conceptual part of the importance of CSR in business models and the need to measure the impact of that term through social media /twitter analysis is well justified, and that it is an active research in the scientific community but has never been done as proposed in the study. What I believe it lacks:
- A clear objective of what the study is looking for. Yes, the questions to be answered are posed, but they are not clearly answered in the results. I believe that the questions must be preceded by a clear objective of the study. For example: To identify the main characteristics of CSR on Twitter.

-The methodology is not clear. It is indicated in the introduction that a machine learning algorithm is going to be used in the research, and then in the methodology it is not even indicated. The methodological phases do not follow the indicated figure. In data mining and network analysis it is not indicated which software is used to perform it. The methodology should give us all the steps to reach the objective, in this case, the concepts that are going to be treated are explained very well but not how it is carried out and why. In short, it is not understood what is the objective to be achieved with each subsection of the different phases. I recommend defining an indicator in each phase that allows an answer to the question posed as an objective. The methodology needs an important improvement in this aspect and to clarify the data used and which are the fields of the download that become key in the analysis.

Validity of the findings

As for the results, their structure is very confusing and does not make it clear whether the questions are being answered. It is true that the data obtained are adequately discussed, but sometimes this makes the objective of the study unclear.
In the first paragraph, the clarification of some results is repeated. Improve the explanation and analysis. I believe that the lack of a consistent and clear methodology means that the results section is a part full of data resulting from the statistical analyses performed but which do not provide a clear view of what is to be achieved. For example, what does the sentiment analysis give me? Data has been indicated but without a clear result. Likewise, I wonder about trends, which are very important but their implications for the objective of the study are not analyzed. Therefore, the results need to be better written and associated to the established objectives.
All this makes the conclusions of the study not good. Instead of concluding the results obtained, the results are repeated, and this should not be done. The conclusions should be revised and provide information on the main characteristics of CSR in social media/Twitter. It is correct to explain the limitations of the study and propose new ways of study, improving the system of indicators or analyzing other social media.

Reviewer 2 ·

Basic reporting

I have a few small remarks.

About the data availability: I could not find where to download data. According to Twitter's policy, for the reproducibility of academic research, the list of tweet ids can (and should) be shared: https://developer.twitter.com/en/developer-terms/more-on-restricted-use-cases.

Regarding literature references, it is quite weird the sentence (line 100) "The concept of corporate social responsibility dates back to the 1920s", followed by an article dating 2007: it would be clearer either to cite the original paper from 1920 or to cite the context, as described in the article of 2007 by Hoffman.

Somewhere there are small typos (I mean, you will always find small typos even in re-reading the paper a billions of times, so I am not considering this as a strong issue, but justas a genuine contribution to the readability of the manuscript :) ): I found two open parenthesis at line 100 and at line 171 the sentence "The social networks currently *use* approximately 4.6 billion *users*", that could be rewritten more efficiently. Moreover, I guess that at line 219 it should be "(Deeva, 2019)".

Figures are nice.The only one that could be improved is Fig.3, since the percentages of negative sentiment are either hard to read (black over dark blue background) or fall outside the bar. I suggest moving the percentage a little on the right and use white fonts.

Experimental design

The only criticism that I can raise regarding the entire analysis is the choice of the keywords for downloading data. I mean, it is quite obvious that when using #csr as a keyword, you will only get data about companies advertising their contribution to society, if any. Otherwise stated, in my view the issue could be that it is not clear, from the selected data, what is the impact of CSR topics respect to the baseline of the standard narrative of the various firms.
To be clearer: it is not an issue per se. Such a choice can be done in order to characterize CSR on its own, without any comparison with other topics. It just have to be justified and emphasized a little more.

Then, it is not clear how the hashtag network is defined and therefore this part lack of clarity.
In fact, the following sentence is quite weird: "A modified Louvain algorithm was used to create clusters using the links between text values" (line 217). Louvain is a community *detection* algorithm, i.e. given a network, it finds its organization in terms of communities. In this sense, it is not creating anything. Moreover, the jargon used to describe the modification proposed by (Deeva, 2019) is quite arduous (I fear it depends on the lack of definition of the graph of hashtags) and hard to understand. It seems (but please, correct me if I am wrong) that what you did is maximising the modularity (this is not what is written at line 220), as defined for the weighted network of hashtags. I guess that the weighted network of hashtags, then, is the network of co-occurrences of hashtags in messages, i.e. the weight between hashtag A and hashtag B is the number of tweets they both appeared in. Please, rephrase the entire paragraph, since it is far from being clear.

At line 227, it seems you are using GePhy. If so, please mention it explicitly.

Validity of the findings

As I mentioned at point 1., I could not find raw data. Regarding the rest, conclusions are well stated and justified.

Additional comments

I found the present manuscript pretty interesting. I believe it is worth to be published, once some of the steps of its analyses are clarified and justified more in detail.

---

## Round 0.2 · accepted · Accept

I am pleased to inform you that your manuscript entitled "Global analysis of Twitter communication in corporate social responsibility area: sustainability, climate change, and waste management" has been accepted for publication in PeerJ Computer Science. Congratulations on your accomplishment!
We appreciate the effort and time you have invested in the revisions and the comprehensive and detailed response to the reviewers' comments. Your answers have helped to clarify and improve the manuscript, and we are confident that the final version will be a valuable contribution to the scientific literature.
Thank you again for your contribution to the journal, and we look forward to publishing your work.
Best regards,

José M. Galán

Reviewer 1 ·

Basic reporting

It is correct, the objective of the study is clearer, it would not be superfluous to include for the benefit of whom the research is carried out (further included)

Experimental design

Improved and correct, as requested in the first revision.

Validity of the findings

The results obtained are interesting for both the business community and the scientific community due to the methodology applied.